# Seasonal Patterns and Species Variability in the Leaf Traits of Dominant Plants in the Tropical Rainforests of Hainan Island, China

**Ruijing Xu** [1,2] , **Quan Qiu** [3] , **Junqing Nong** [2], **Shaohui Fan** [2] and **Guanglu Liu** [2,*]

1   International Center for Bamboo and Rattan Sanya Research Base, Sanya 572000, China
2   Key Laboratory of National Forestry and Grassland Administration, Beijing for Bamboo & Rattan Science and Technology, International Center for Bamboo and Rattan, Beijing 100102, China
3   Guangdong Key Laboratory for Innovative Development and Utilization of Forest Plant Germplasm, College of Forestry and Landscape Architecture, South China Agricultural University, Guangzhou 510642, China
*   Correspondence: liuguanglu@icbr.ac.cn; Tel.: +86-10-84789848

**Abstract:** The leaf traits measured in multiple species are known to vary between seasons, but there is a knowledge gap relating to the seasonal variability and environmental adaptation of plants in tropical rainforests. To investigate the dynamics of the functional traits of dominant species in tropical rainforests and the differences in their adaptation strategies to seasonal drought, the results of this study can provide a scientific basis for tropical rainforest conservation resource protection. Six dominant species, including three trees (*Hopea reticulata*, *Vatica mangachapoi*, and *Diospyros chunii*) and three vine plants (*Ancistrocladus tectorius*, *Phanera khasiana*, and *Uvaria sanyaensis*), in tropical lowland rainforest in the Ganzaling Nature Reserve of Hainan province were selected as study objectives. The key leaf traits were studied using the paraffin section method, leaf epidermis segregation method, and Li-6400 portable photosynthesis system in June, September, December, 2019, and March, 2020. Results showed that significant differences in photosynthetic physiology and morphological and structural parameters among species, as well as seasonal variability, were observed in leaf photosynthetic physiology, but not in leaf morphological or structural parameters. A phenotypic plasticity index (PPI) analysis revealed more variability in leaf photosynthetic physiology (Average PPI = 0.37) than in leaf anatomical structure and morphology (Average PPI = 0.26), suggesting that they adapt to seasonal changes primarily by regulating photosynthetic physiological parameters rather than leaf morphology or anatomical structure. The dominant trees were found to have higher water use efficiency, leaf dry-matter content, and smaller leaf areas compared to vine plants. This indicates that the dominant tree species depend on high water use efficiency and leaf morphological characteristics to adapt to seasonal changes. The majority of leaf anatomical structure parameters associated with drought tolerance were higher in the three dominant vine species, indicating that the dominant vine species adapted to drought stress primarily by altering the leaf anatomical structure This study provides information on how tropical rainforest plants adapt to seasonal drought as well as supporting the protection of tropical rainforest ecosystems.

**Keywords:** tropical rainforests; leaf trait; seasonal drought; leaf anatomical structure; photosynthesis

## 1. Introduction

Global forest health is at risk from climate change as a result of drought and heat-induced tree mortality [1]. Drought—induced by climate change—has received particular attention due to its potential threat to various ecosystems. Previous studies have suggested that aridity drives ecosystem thresholds worldwide [1,2], and Amazon carbon sinks decline after droughts; increased evaporative demand is caused by increasing temperature due

to climate change [3]. It is widely recognized that global ecosystems, including tropical rainforests, are facing drought risks. As a result, this poses a great challenge to the conservation of rainforests. The photosynthetic and hydraulic characteristics of forests influence resistance and resilience to drought stress across a wide range of ecosystems, and the diversity of functional traits has been shown to affect forests' resilience to drought stress [4,5]. Thus, preserving the diversity of the functional traits of rainforests is important. Prior to that, it is crucial to understand the differences in functional characteristics among tree species.

Leaves play a crucial role in the exchange of carbon, water, and energy between the atmosphere and land [6]. To adjust to different habitats, plants show different anatomical traits, leaf photosynthetic characteristics, and morphological traits [7,8]; leaf functional traits (such as SL, SD, and SLA) are adaptive traits in tree species and characterize the population/species drought resistance and/or water retention ability [9]. It is widely known that leaf traits are capable of characterizing plant function, and therefore are widely used to evaluate a plant's response to differing environmental conditions, such as drought, shade, and nutrient deficiency. In response to seasonal changes or stress, photosynthetic physiological factors such as leaf water potential, transpiration rate, and water use efficiency were regulated, as well as structural traits (e.g., anatomical structure) and morphological traits (e.g., surface area). In previous studies, leaf traits varying between seasons was widely measured in multiple species and various forests across a variety of regions [6,8,10–13]. These seasonal variations are commonly attributed to changes in the environment conditions, and each functional trait responds to the changing conditions differently. For instance, *Populus euphratica* exhibits seasonal fluctuations and temperature dependence on photosynthetic parameters and stomatal conductance at the leaf scale [13]; Amazonian forests exhibit seasonal and drought-related changes in leaf area profiles influenced by height and light conditions [12].

Seasonal drought has a significant impact on plant growth, water physiology, nutrient cycling, and gene expression [14–18]. For example, seasonal drought reduces the physiological functions of the leaves of tropical tree species in Panama and affects the stomatal conductance of tropical forest leaves, which in turn reduces the photosynthesis and growth rate of plants [18]; seasonal drought also leads to a reduction in soil moisture, which affects transpiration in subtropical coniferous forests [16]. Seasonal drought is considered one of the most important factors determining the plant communities on Hainan Island's tropical lowland rainforest ecosystem [19]. The dominant species, whether they are trees, shrubs, or vines, should be able to cope with seasonal drought. An earlier study suggested that bamboo species from this site adjust their leaf traits to seasonal drought conditions [20]. It is also possible for dominant trees and vines to adapt their leaf traits to seasonal drought, yet it is unclear that there are variations among species in leaf trait-based response mechanisms.

In this study, six dominant species, including three trees (*Hopea reticulata*, *Vatica mangachapoi*, and *Diospyros chunii*) and three vine plants (*Ancistrocladus tectorius*, *Phanera khasiana*, and *Uvaria sanyaensis*), were selected to determine their key leaf traits, including gas, morphological, and structural parameters, thus fostering analysis of their adverse adaption strategy to seasonal drought. We hypothesized that (1) there are species and seasonal differences in leaf functional traits (photosynthetic physiology and morphological and structural parameters) of dominant species in tropical rainforests; (2) photosynthetic physiological parameters are the important leaf trait variations through which dominant species adapt to seasonal changes in their environment; (3) there are differences in environmental adaptation strategies between trees and vines. The findings could contribute to the understanding of the survival and adaptation strategies of tropical plant species in rainforests, as well as provide a scientific basis for the conservation and use of rainforest plant resources.

## 2. Materials and Methods

### 2.1. Study Site

The study site (109°40′4″ E, 18°23′2″ N, altitude 202 m; Figure 1) is located in Ganzaling Nature Reserve on Hainan Island, China, which is located at the junction of Sanya City and the southern part of Baoting County. It is a lowland hilly landform, with a slope < 50° and an altitude of 50–681 m. The soil parent material is granite, the rock exposure rate is 10%, and the sand content is approximately 20%. The study area experiences a tropical marine monsoon climate with an annual rainfall of 1200–1800 mm and alternating wet and dry seasons. Over 90% of the rainfall occurs during the wet season, from May to November, and the dry season from December to May of the following year experiences low rainfall (Figure 2). The average annual temperature is 24 °C. Tropical lowland secondary rainforests originally covered the area. In the different vegetation layers, the dominant species are summarized as follows: the dominant tree species include *H. reticulata*, *D. chunii*, *V. mangachapoi*, etc.; the dominant shrub species include *Licuala spinosa*, *Ixora hainanensis*, *Ardisia lindleyana*, etc.; the dominant herbaceous species include *Scleria terrestris*, *Blechnum orientale*, *Alpinia oblongifolia*, etc.; the dominant vine species include *A. tectorius*, *P. khasiana*, *U. sanyaensis*, palm vines, etc.

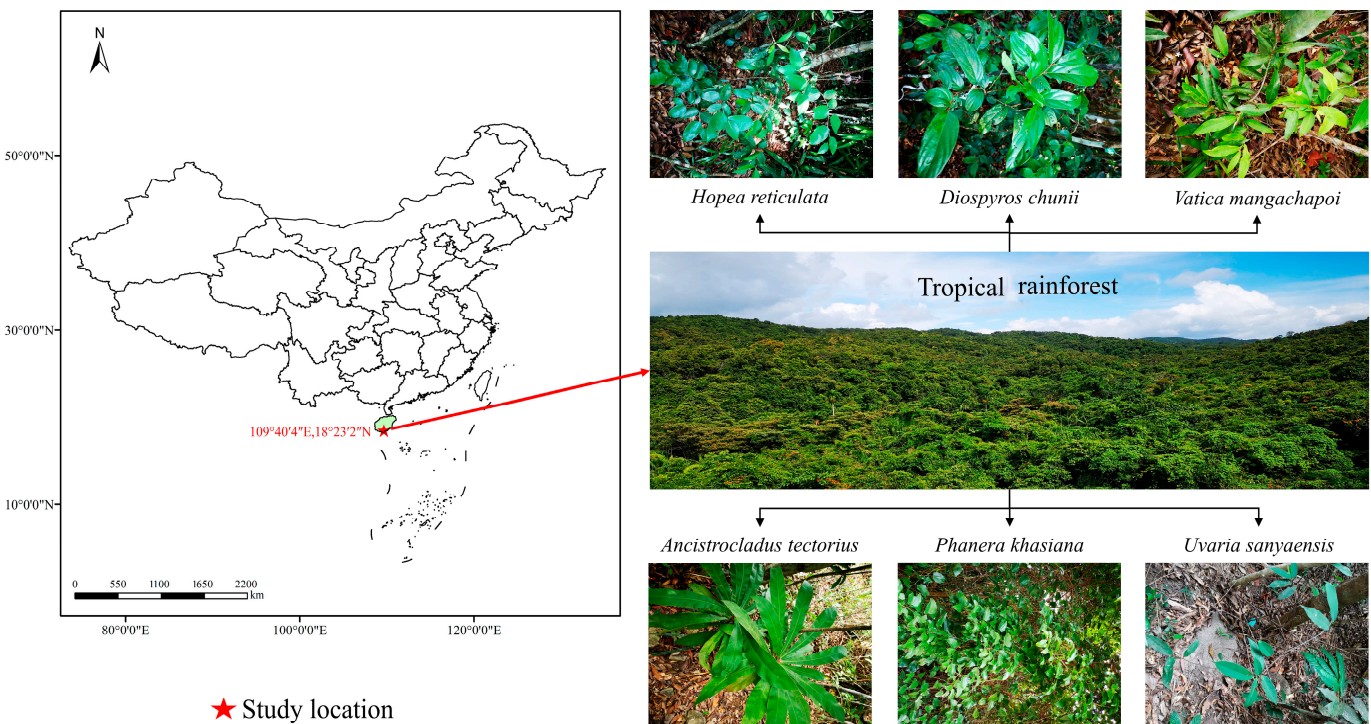

**Figure 1.** Study site and materials.

### 2.2. Materials

Six dominant species, including three trees (*H. reticulata*, *V. mangachapoi*, and *D. chunii*) and three vine plants (*A. tectorius*, *P. khasiana,* and *U. sanyaensis*), were selected as study objectives (Figure 1). *H. reticulata*, a Grade II national protected plant, is found only in Ganzaling (our experimental site) in China as well as Vietnam. Mature *H. reticulata* trees can reach a height of 15 m. *V. mangachapoi,* a Grade II national protected plant, is an indicator species in tropical rainforests. The mature *V. mangachapoi* tree can reach a height of approximately 20 m. *D. chunii* is an endemic species native to China, found only in Sanya, and grows to a height of 4–7 m. *A. tectorius is* a moderately shade -loving species with long hook-shaped structures that can grow up to 10 m. A sun-loving species, *P. khasiana*, climbs up to the canopy of the rainforest. It is over 20 m in length. *U. sanyaensis* is a shade-loving species that grows in the understory and measures only 5 m in length.

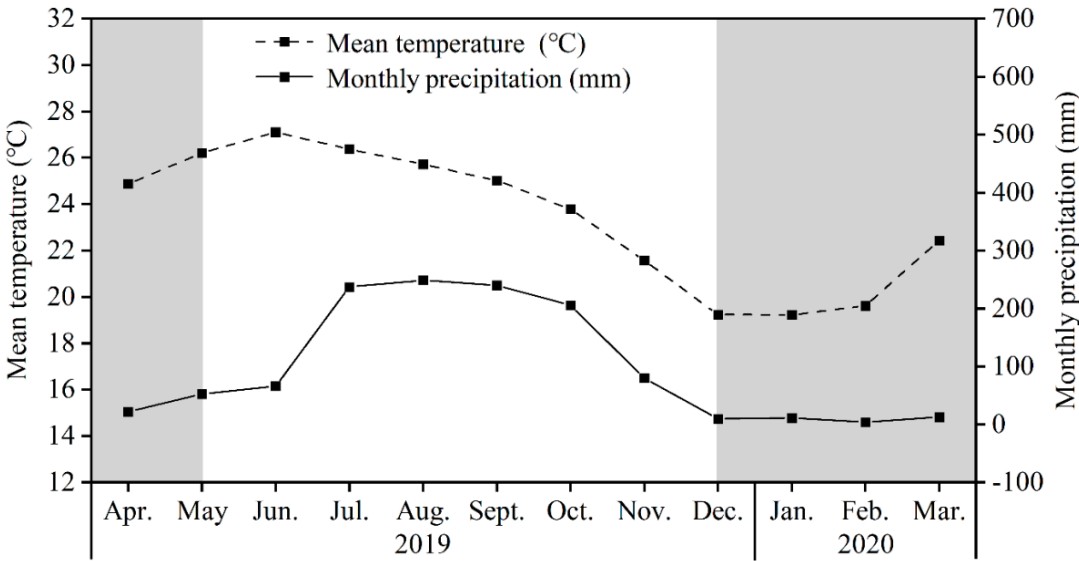

**Figure 2.** Monthly changes of precipitation and mean temperature in Sanya city (April 2019–March 2020).

### 2.3. Experimental Design and Sampling

Before experiments, 3–5 individuals with similar growth status per species were selected and marked as sample plants. They were well protected during the experiments. Leaf samples were collected from these marked plants in mid-June, mid-September, and mid-December 2019 and March 2020 for leaf trait measurements (Table 1). Photosynthetic parameters were measured on a sunny day, and the youngest branch at 1.6 m above ground level was selected at five locations in the canopy: east, west, south, north, and center; the developed mature leaf (penultimate 4th leaf at the end of the branch) was selected on the branch.

**Table 1.** Symbols and abbreviations for leaf traits and their measurement information.

| Abbreviation | Description | Unit | Measurement (Time; Sample no.) |
|---|---|---|---|
| Pn | Net photosynthetic rate | $\mu mol\ m^{-2}\ s^{-1}$ | Jun., Sep., Dec., Mar.; 5 |
| Gs | Stomatal conductance | $\mu mol\ m^{-2}\ s^{-1}$ | Jun., Sep., Dec., Mar.; 5 |
| Ci | Intercellular carbon dioxide concentration | $\mu mol\ mol^{-1}$ | Jun., Sep., Dec., Mar.; 5 |
| Tr | Transpiration rate | $\mu mol\ m^{-2}\ s^{-1}$ | Jun., Sep., Dec., Mar.; 5 |
| WUE | Water use efficiency | $\mu mol\ mmol^{-1}$ | Jun., Sep., Dec., Mar.; 5 |
| Ls | Stomatal limitation | / | Jun., Sep., Dec., Mar.; 5 |
| VPD | Vapor pressure deficit | KPa | Jun., Sep., Dec., Mar.; 5 |
| AQE | Apparent quantum efficiency | $mol\ mol^{-1}$ | Jun., Sep., Dec., Mar.; 5 |
| Pnmax | Maximum photosynthetic rates | $\mu mol\ m^{-2}\ s^{-1}$ | Jun., Sep., Dec., Mar.; 5 |
| LSP | Light saturation point | $\mu mol\ m^{-2}\ s^{-1}$ | Jun., Sep., Dec., Mar.; 5 |
| LCP | Light compensation point | $\mu mol\ m^{-2}\ s^{-1}$ | Jun., Sep., Dec., Mar.; 5 |
| Rd | Dark respiration efficiency | $\mu mol\ m^{-2}\ s^{-1}$ | Jun., Sep., Dec., Mar.; 5 |
| LT | leaf thickness | $\mu m$ | Jun., Sep., Dec., Mar.; 5 |
| USCT | Upper stratum corneum thickness | $\mu m$ | Jun., Sep., Dec., Mar.; 5 |
| UET | Upper epidermal thickness | $\mu m$ | Jun., Sep., Dec., Mar.; 5 |
| LET | Lower epidermal thickness | $\mu m$ | Jun., Sep., Dec., Mar.; 5 |
| PTT | Palisade tissue thickness | $\mu m$ | Jun., Sep., Dec., Mar.; 5 |
| STT | Spongy tissue thickness | $\mu m$ | Jun., Sep., Dec., Mar.; 5 |

**Table 1.** *Cont.*

| Abbreviation | Description | Unit | Measurement (Time; Sample no.) |
|---|---|---|---|
| SAFVB | Sectional area of first-vascular bundle | $mm^2$ | Jun., Sep., Dec., Mar.; 5 |
| DSVB | Diameter of second-order vascular bundle | μm | Jun., Sep., Dec., Mar.; 5 |
| SASVB | Sectional area of second-order vascular bundle | $mm^2$ | Jun., Sep., Dec., Mar.; 5 |
| DBAVB | Distance between adjacent vascular bundle | μm | Jun., Sep., Dec., Mar.; 5 |
| SL | Stomata length | μm | Jun., Sep., Dec., Mar.; 5 |
| SW | Stomata width | μm | Jun., Sep., Dec., Mar.; 5 |
| AS | Area of single stomata | $μm^2$ | Jun., Sep., Dec., Mar.; 5 |
| SD | Stomata density | number $mm^{-2}$ | Jun., Sep., Dec., Mar.; 5 |
| AP | Percent of stomata area | % | Jun., Sep., Dec., Mar.; 5 |
| LA | Leaf area | $cm^2$ | Jun., Sep., Dec., Mar. 3 |
| LDMC | Leaf dry-matter content | $g\,kg^{-1}$ | Jun., Sep., Dec., Mar.; 3 |
| SLA | Specific leaf area | $m^2\,kg^{-1}$ | Jun., Sep., Dec., Mar.; 3 |

*2.4. Leaf Traits Measurements*

2.4.1. Gas Exchange Parameter Measurements

Leaf photosynthesis capability was examined by measuring the instantaneous values of the gas exchange parameter. Net photosynthetic rate (Pn), stomatal conductance (Gs), intercellular $CO_2$ concentration (Ci), and transpiration rate (Tr) were determined on the 3–5th fully expanded leaves (from the apex) in mid-June, mid-September, and mid-December, 2019, and March, 2020. Pn values were recorded using an LI-6400 portable photosynthesis system (Li-Cor Inc., Lincoln, NE, USA). To obtain stable measurements and simulate actual external environmental conditions, in accordance with actual light conditions under different irradiance treatments, photosynthetic photon flux density (PPFD) at the leaf surface was set at 1000 μmol $m^{-2}$ $s^{-1}$. Water use efficiency (WUE) was calculated as WUE = Pn/Tr. The meteorological data were recorded with the LI-6400 portable photosynthesis system, including air temperature (*Ta*; °C), air relative humidity (*RH*; %), and ambient $CO_2$ concentration (Ca). The stomatal limitation value (Ls) was then calculated using the following formula: Ls = 1 − –Ci/Ca. The water vapor pressure deficit (VPD; kPa) was calculated according to the formula [21]:

$$VPD = 0.611 \times \exp\left(\frac{17.27 \times Ta}{Ta + 237.3}\right) \times (1 - RH)$$

To further assess leaf photosynthesis capabilities and shade tolerance, the light response curve was established, and then these data were analyzed to calculate apparent quantum efficiency (AQE), maximum photosynthetic rates (Pnmax), light saturation point (LSP), light compensation point (LCP), and dark respiration efficiency ($R_d$) using a photosynthesis calculation software (version 4.11) based on a non-rectangular hyperbolic model [22]. In this experiment, the PPFD was set to the following gradients: 2400, 2000, 1800, 1600, 1400, 1200, 1000, 800, 500, 200, 50, and 0 μmol $m^{-2}$ $s^{-1}$, and the $CO_2$ concentration was set to 400 μmol $mol^{-1}$.

2.4.2. Morphological Determination

Leaf dry matter content is an index of mass investment in photosynthetic organs that is positively correlated with leaf density and negatively correlated with plant growth [23,24]. Specific leaf area (SLA, the ratio of leaf area to leaf dry mass) and leaf dry matter content (LDMC, the ratio of leaf dry mass to fresh mass) of mature leaves was measured to analyze their adjustments in organic matter and energy partitioning to the varying environments. Leaf area was determined by scanning the leaves with an Epson perfection V19 scanner and calculating the area in the leaf area calculation program (version 1.1); leaf dry matter content was calculated by cutting off the petioles of mature leaves, soaking them in water for 12 h, absorbing the water from the leaf surface with absorbent paper, and weighing the

saturated fresh weight, then killing at 105 °C and drying at 80 °C to constant weight, and weighing the dry weight. LDMC = dry weight (g)/saturated fresh weight (kg); SLA = leaf area (m$^2$)/dry weight (kg).

### 2.4.3. Anatomical Measurements

Paraffin sections of the leaves were prepared for anatomical measurements, and the material was softened, de-watered and immersed in wax, embedded, sectioned, stained with red-solid green, and sealed with neutral resin. The sections were observed and photographed under an Osmia PH50-3M100 light microscope, and each tissue structure index was measured using ImageView image analysis software. The following indices were measured: leaf thickness (LT), upper cuticle thickness (USCT), papillae thickness (MPT), upper epidermal thickness (UET), lower epidermal thickness (LET), number of vesicular cells (NBC), cross-sectional area of vesicular cells (SABC), cross-sectional area of single spindle cells (SAFC), cross-sectional area of primary vascular bundles (SAFVB), diameter of secondary vascular bundles (DSVB), and 12 secondary vascular bundle cross-sectional area (SASVB), and secondary vascular bundle spacing (DBAVB). For each species, 30 statistical index values were taken in each observation.

Leaf epidermal sections were prepared through the maceration method using glacial acetic acid and 30% hydrogen peroxide. Leaf tissues were placed in a decoction bottle containing glacial acetic acid and 30% hydrogen peroxide, and they were then incubated for 27 h at 60 °C. Following the separation of the upper and lower epidermis and leaf tissues, the dissociated material was removed and transferred to distilled water. After splitting the upper and lower epidermis with a brush, the washed leaf epidermis was stained with 1% fenugreek and 50% alcohol solution for 5 min, and the slices ware then sealed with neutral gum. The slices were observed and photographed under an Osmia PH50-3M100 light microscope, and each tissue structure index was measured using ImageView image analysis software. The stomata length (SL), stomata width (SW), and area of single stomata (AS) were determined. The stomata density (SD) and percent of stomata area (AP) were calculated, where SD = number of stomata in the field of view/area of view, and AP = AS × SD × 100%.

### 2.5. Statistical Analyses

The phenotypic plasticity index (PPI = (max − min)/max, where max and min represent mean maximum and minimum values for each leaf trait, respectively) was calculated separately for each morphological and physiological trait [25]. Differences in plant traits between seasons and species were evaluated using a Kruskal–Wallis test, followed by pairwise multiple comparisons. All statistical analyses were conducted using SPSS software (ver. 20.0; SPSS Inc., Chicago, IL, USA). Figures were constructed using OriginPro software (Ver. 2021. OriginLab Corporation, Northampton, MA, USA).

## 3. Results

### 3.1. Gas Exchange Parameters Analysis

3.1.1. Instantaneous Gas Exchange Parameter Analysis

A seasonal analysis of gas exchange parameters for six species is shown in Figure 3. The highest levels of Pn were observed in the rainy season (June or September) for *H. reticulata*, *V. mangachapoi*, *P. khasiana*, and *U. sanyaensis*, whereas the highest levels were recorded in December for *D. chunii* and *A. tectorius*. For all six species, Gs, Ci, Tr, and VPD were highest during the rainy season (from June or September); however, their WUE and Ls had reversed patterns, with the highest values occurring during the dry season (December or March). These finding suggesting that the gas exchange parameters such as Pn, Gs, Ci, and Tr of dominant species in tropical rainforests were higher during the rainy season, but their WUE performed better in the dry season.

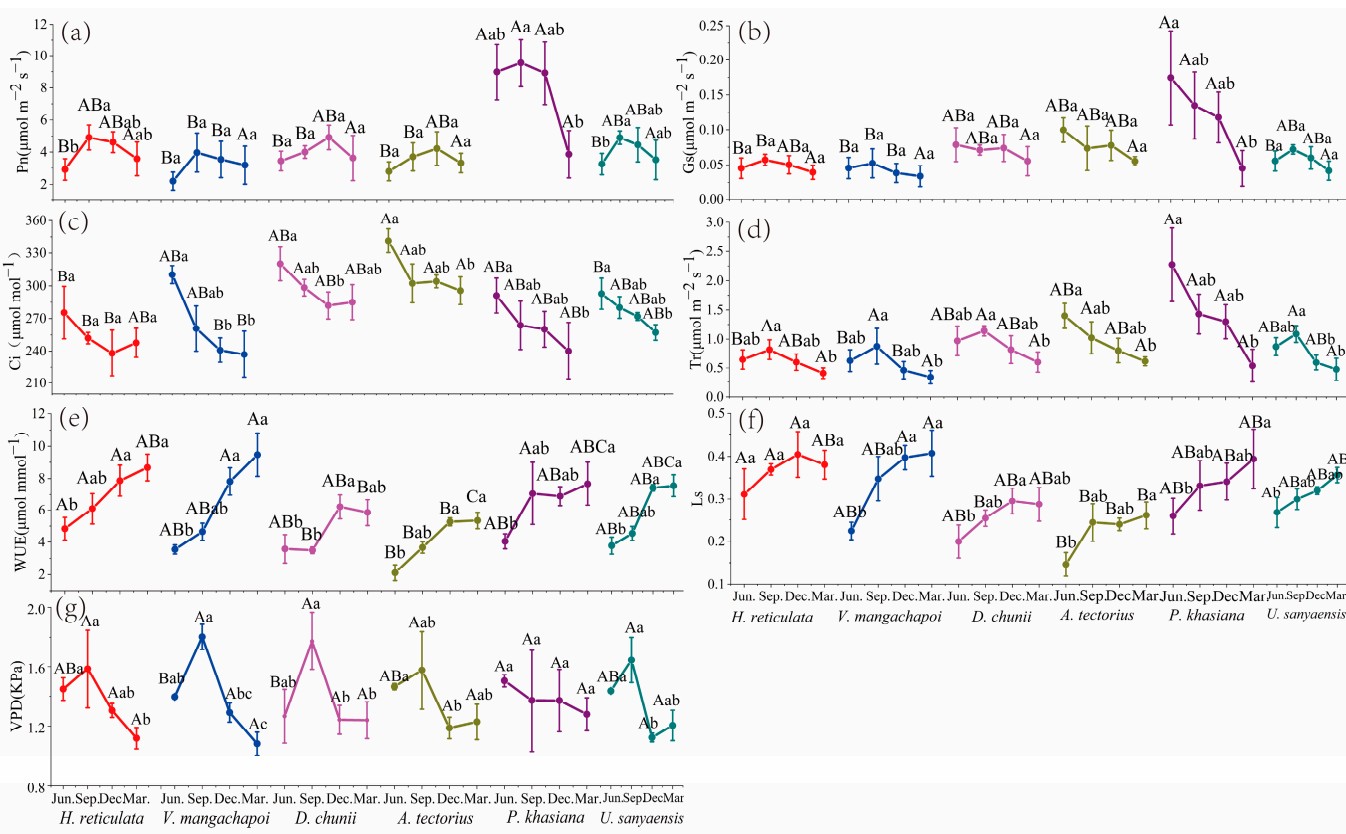

**Figure 3.** Gas exchange parameters of six species in different months. (**a**) Pn; (**b**) Gs; (**c**) Ci; (**d**) Tr; (**e**) WUE; (**f**) Ls; (**g**) VPD. Different capital letters for the same month indicate significant differences between species at the 0.05 level, while different lowercase letters for the same species indicate significant differences between months at the 0.05 level ($p < 0.05$). See Table 1 for full explanations of the abbreviations.

There was a wide variation among species regarding gas exchange parameters, with *P. Khasiana* generally having higher levels of Pn and Gs than other species, whereas *V. Mangachapoi* performed at the lowest level, demonstrating that *P. Khasiana* had a stronger photosynthetic capacity, while *V. Mangachapoi* showed a weaker capability. Additionally, several interspecific and monthly differences in gas exchange parameters of dominant tropical lowland rainforest species are evident for the other four species.

Considering the comparison of plant life types, the dominant tree species had higher WUE ($6.01 \pm 2.16$ µmol mmol$^{-1}$), Ls ($0.32 \pm 0.08$), and VPD ($1.38 \pm 0.26$ KPa) than the dominant vine species ($5.45 \pm 1.97$ µmol mmol$^{-1}$; $0.29 \pm 0.07$; $1.37 \pm 0.22$ KPa), but their Pn ($3.74 \pm 1.21$ µmol m$^{-2}$ s$^{-1}$), Gs ($0.05 \pm 0.02$ µmol m$^{-2}$ s$^{-1}$), Ci ($270.65 \pm 32.20$ µmol mol$^{-1}$), and Tr ($0.69 \pm 0.30$ µmol m$^{-2}$ s$^{-1}$) were all lower than those of the dominant vine species ($5.12 \pm 2.69$ µmol m$^{-2}$ s$^{-1}$; $0.08 \pm 0.05$ µmol m$^{-2}$ s$^{-1}$; $284.57 \pm 29.99$ µmol mol$^{-1}$; $1.03 \pm 0.57$ µmol m$^{-2}$ s$^{-1}$). Based on these findings, tree species exhibited a higher water use capacity than vine species but exhibited a lower level of photosynthetic capacity.

### 3.1.2. Light Response Curve Analysis

A light response curve analysis for the six species is shown in Figure 4. The AQE and Pnmax of *H. reticulata*, *V. mangachapoi*, *A. tectorius*, *P. khasiana*, and *U. sanyaensis* all reached their maximum values during the rainy season (June or September), while their minimum values occurred in different months. The AQE and Pnmax of these five species generally showed insignificant differences between months ($p > 0.05$), except for the Pnmax of *P. khasiana*. The maximum and minimum values of LSP of *H. reticulata*, *D. chunii*, and *U. sanyaensis* occurred in the rainy season, while the opposite trend was observed for

*P. khasiana*, whose maximum and minimum values occurred in the dry season. There were significant monthly differences in the LSP of *H. reticulata*, *A. tectorius*, and *P. khasiana* ($p < 0.05$). The maximum values of LCP and Rd of *H. reticulata* occurred in March and the minimum values in June, while *V. mangachapoi*'s values were reversed; *A. tectorius*'s maximum and minimum values occurred in June and September, respectively ($p > 0.05$); *P. khasiana* and *U. sanyaensis* showed maximum values during the rainy season and minimum values during December, and the differences were not statistically significant ($p > 0.05$). In the comparison of different species, the AQE, Pnmax, LSP, LCP, and Rd of *P. khasiana* were the highest, except for LSP in March, and the other species performed differently in different months, suggesting that *P. khasiana* had an excellent photosynthesis ability, and the other species exhibited some monthly differences in the fitted characteristic parameters. There was no significant difference in AQE between species ($p > 0.05$) over the four months, except for the AQE of *U. sanyaensis* in December; the Pnmax of *P. khasiana* and *V. mangachapoi* were significantly different in September and December ($p < 0.05$). Except for LCP in March, there were significant differences between LCP, Rd max and min of the six species ($p < 0.05$). A comparison of plant life types revealed that Pnmax ($5.30 \pm 2.62$ µmol m$^{-2}$ s$^{-1}$), LSP ($967.07 \pm 348.10$ µmol m$^{-2}$ s$^{-1}$), LCP ($7.14 \pm 6.10$ µmol m$^{-2}$ s$^{-1}$), and Rd ($0.44 \pm 0.46$ µmol m$^{-2}$ s$^{-1}$) were greater in the dominant vine species than in the dominant tree species ($3.99 \pm 1.17$ µmol m$^{-2}$ s$^{-1}$, $718.82 \pm 208.97$ µmol m$^{-2}$ s$^{-1}$, $4.58 \pm 3.63$ µmol m$^{-2}$ s$^{-1}$, $0.24 \pm 0.18$ µmol m$^{-2}$ s$^{-1}$), and their mean AQE levels were similar.

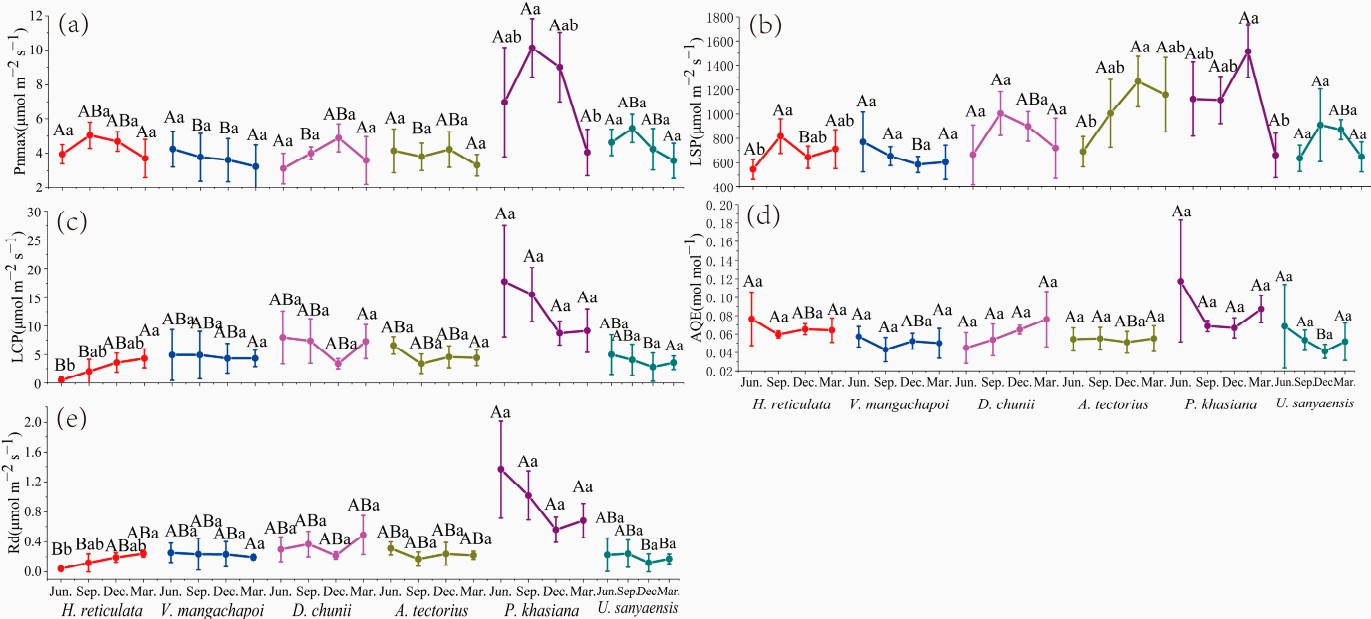

**Figure 4.** Characteristic parameters of the light response curve of the six species in different months. (**a**) Pnmax; (**b**) LSP; (**c**) LCP; (**d**) AQE; (**e**) Rd. Different capital letters for the same month indicate significant differences between species at the 0.05 level, while different lowercase letters for the same species indicate significant differences between months at the 0.05 level ($p < 0.05$). See Table 1 for full explanations of the abbreviations.

## 3.2. Leaf Anatomical Structure Analysis

Comparing different months (Table S1), LT, USCT, UET, LET, PTT, STT, DSVB, SASVB, and AP were not significantly different in all four months ($p > 0.05$) for all species, suggesting that these indicators displayed more stability over time. SAFVB, SL, and SD differed significantly between months in *A. tectorius* ($p < 0.05$), but there were no significant monthly differences between other species ($p > 0.05$). AS was the indicator with the greatest monthly variation between all indicators, with significant monthly differences ($p < 0.05$) in *V. mangachapoi*, *D. chunii*, and *U. sanyaensis*. The above results demonstrated that the leaf

epidermal, fleshy, and vascular tissues differed less between months, and there were some monthly differences in stomata, but the monthly differences were not significant in most of the indicators.

Comparing different species (Table S1), the leaf tissue structure had large interspecific differences. *U. sanyaensis* and *A. tectorius* had thicker LT, while *V. mangachapoi* and *P. khasiana* had thinner LT, with significant differences between the maximum and minimum LT values of different species ($p < 0.05$); both UET and LET were highest in *U. sanyaensis*. UET and LET were both highest in *U. sanyaensis* (except for UET in December) and lowest in *A. tectorius* or *V. mangachapoi*, with significant differences between their maximum and minimum values ($p < 0.05$). The PTT and STT of each species were highest for *P. khasiana* and *A. tectorius*, respectively. Except for SAFVB in December, SAFVB, DSVB, and SASVB were all highest in *A. tectorius*, reflecting that *A. tectorius* has a strong vascular tissue system. Except for SW in September and March, SL, SW, and AS were highest in *H. reticulata* and lowest in *P. khasiana*, and SL and AS were significantly higher in *H. reticulata* than in *P. khasiana* ($p < 0.05$). SD and AP were both significantly higher in *P. khasiana* than in *A. tectorius* and *D. chunii* in September ($p < 0.05$). The USCT and DBAVB of different species behaved differently in each month. The above results reflected that *U. sanyaensis* had well-developed epidermal tissue, *P. khasiana* and *A. tectorius* had thicker chloroplast tissue, *A. tectorius* also had a well-developed vascular system, *H. reticulata* had larger stomata, *D. chunii* and *A. tectorius* had smaller stomata, and *P. khasiana* had denser stomata.

### *3.3. Leaf Morphology*

Comparing different months (Figure 5), except for *D. chunii*'s SLA in September, the LA and SLA of *H. reticulata*, *V. mangachapoi*, *D. chunii*, *A. tectorius*, and *P. khasiana* were all highest during the rainy season (June or September) and lowest during the dry season (December or March), but *U. sanyaensis*'s values were reversed. The LDMC values of all six species were the lowest in June and the highest during the dry season (except for *V. mangachapoi*), and the LDMC values of all species (except for *P. khasiana*) were not significantly different between months ($p > 0.05$). These results indicated that leaf morphological traits differed between months, but most of the differences were not significant. Comparing different species (Figure 5), LA was highest in *A. tectorius* and lowest in *H. reticulata* in all four months, with significant differences between them in September and December ($p < 0.05$). The LDMC of *V. mangachapoi* was higher than the remaining five species, except in June, and the difference in LDMC between *V. mangachapoi* and the remaining five species was significant ($p < 0.05$). SLA values were highest in *A. tectorius* in different months, and the lowest values of each species showed different performance in different months. Comparing different life types, the LDMC of tree layer dominant species ($404.31 \pm 93.36$ g kg$^{-1}$) was higher than that of vine dominant species ($359.92 \pm 73.51$ g kg$^{-1}$), but its LA ($24.76 \pm 8.15$ cm$^2$) and SLA ($1081.18 \pm 373.87$ m$^2$ kg$^{-1}$) were lower than that of vine dominant species ($112.76 \pm 91.52$ cm$^2$, $3148.43 \pm 2188.08$ m$^2$ kg$^{-1}$).

### *3.4. Phenotypic Plasticity of Leaf Traits*

The results of PPI of 30 leaf traits are shown in Table 2. Tr, WUE, LCP, and Rd averaged PPIs above 0.50 for all six species, and average PPIs for photosynthetic physiological characteristics parameters (0.44) were higher than those for morphological structural parameters (0.25), reflecting that the dominant species of tropical rainforests had higher plasticity of photosynthetic physiological characteristics parameters such as Tr, WUE, LCP, and Rd, and were more adaptable to the environment and had more stable morphological structural parameters. Among the six species, *P. khasiana* and *A. tectorius* exhibited higher average PPIs (0.35–0.37) due to their extremely high PPI for photosynthetic physiology (0.49) and morphology (0.32), respectively, suggesting that they had stronger environmental adaptability. Moreover, *V. mangachapoi* and *H. reticulata* were found to show relatively weak environmental adjustment capacity for their lower average PPIs (0.29–0.31).

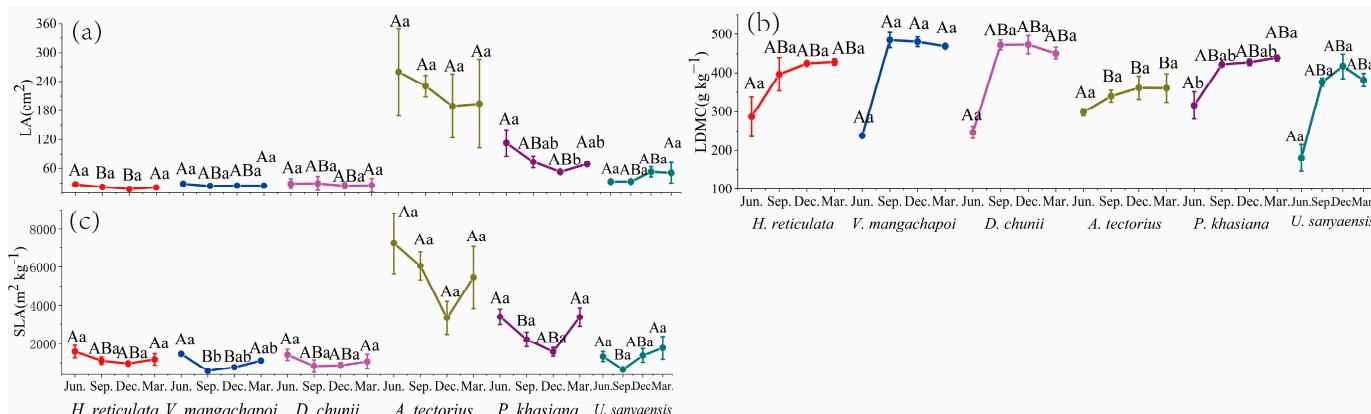

**Figure 5.** Leaf morphological traits of the six species in different months. (**a**) LA; (**b**) LDMC; (**c**) SLA. Different capital letters for the same month indicate significant differences between species at the 0.05 level, while different lowercase letters for the same species indicate significant differences between months at the 0.05 level ($p < 0.05$). See Table 1 for full explanations of the abbreviations.

**Table 2.** Seasonal phenotypic plasticity index in leaf traits of the six dominant plants.

|  | Index | *Hopea reticulata* | *Vatica man-gachapoi* | *Diospyros chunii* | *Ancistrocladus tectorius* | *Phanera khasiana* | *Uvaria sanyaensis* |
|---|---|---|---|---|---|---|---|
| Photosynthetic physiology | Pn | 0.40 | 0.45 | 0.30 | 0.34 | 0.60 | 0.33 |
|  | Gs | 0.31 | 0.35 | 0.30 | 0.45 | 0.74 | 0.42 |
|  | Ci | 0.14 | 0.24 | 0.12 | 0.13 | 0.18 | 0.12 |
|  | Tr | 0.50 | 0.61 | 0.47 | 0.56 | 0.76 | 0.56 |
|  | WUE | 0.44 | 0.62 | 0.44 | 0.60 | 0.47 | 0.50 |
|  | Ls | 0.23 | 0.45 | 0.32 | 0.44 | 0.34 | 0.25 |
|  | VPD | 0.29 | 0.40 | 0.30 | 0.25 | 0.15 | 0.32 |
|  | AQE | 0.22 | 0.25 | 0.41 | 0.07 | 0.43 | 0.40 |
|  | Pnmax | 0.26 | 0.24 | 0.36 | 0.21 | 0.60 | 0.34 |
|  | LSP | 0.33 | 0.24 | 0.34 | 0.45 | 0.56 | 0.30 |
|  | LCP | 0.88 | 0.12 | 0.58 | 0.49 | 0.51 | 0.44 |
|  | Rd | 0.84 | 0.23 | 0.56 | 0.46 | 0.59 | 0.51 |
|  | Average | 0.41 | 0.35 | 0.37 | 0.37 | 0.49 | 0.37 |
| Morphology | LT | 0.06 | 0.08 | 0.10 | 0.27 | 0.18 | 0.07 |
|  | USCT | 0.24 | 0.35 | 0.30 | 0.29 | 0.37 | 0.16 |
|  | UET | 0.14 | 0.12 | 0.06 | 0.20 | 0.16 | 0.11 |
|  | LET | 0.08 | 0.12 | 0.12 | 0.11 | 0.24 | 0.14 |
|  | PTT | 0.25 | 0.08 | 0.06 | 0.43 | 0.17 | 0.17 |
|  | STT | 0.01 | 0.12 | 0.22 | 0.35 | 0.40 | 0.44 |
|  | SAFVB | 0.44 | 0.26 | 0.58 | 0.85 | 0.42 | 0.39 |
|  | DSVB | 0.16 | 0.13 | 0.35 | 0.38 | 0.13 | 0.06 |
|  | SASVB | 0.24 | 0.29 | 0.59 | 0.62 | 0.25 | 0.31 |
|  | DBAVB | 0.35 | 0.34 | 0.53 | 0.14 | 0.27 | 0.23 |
|  | SL | 0.09 | 0.15 | 0.16 | 0.17 | 0.13 | 0.12 |
|  | SW | 0.16 | 0.06 | 0.11 | 0.11 | 0.03 | 0.12 |
|  | AS | 0.18 | 0.27 | 0.25 | 0.27 | 0.14 | 0.29 |
|  | SD | 0.04 | 0.04 | 0.07 | 0.16 | 0.06 | 0.10 |
|  | AP | 0.20 | 0.26 | 0.30 | 0.33 | 0.18 | 0.30 |
|  | LA | 0.35 | 0.16 | 0.16 | 0.27 | 0.53 | 0.38 |
|  | LDMC | 0.33 | 0.51 | 0.48 | 0.17 | 0.28 | 0.57 |
|  | SLA | 0.39 | 0.60 | 0.41 | 0.54 | 0.53 | 0.63 |
|  | Average | 0.21 | 0.22 | 0.27 | 0.32 | 0.25 | 0.26 |
| Total Average |  | 0.31 | 0.29 | 0.32 | 0.35 | 0.37 | 0.32 |

*3.5. Principal Component Analysis of Leaf Traits*

Principal component analysis (PCA) showed in Figure 6. The leaf traits of *A. tectorius* and *P. khasiana* differed markedly from those of the other five species, with *A. tectorius* being unique in its special leaf anatomical traits and *P. khasiana* being unique in its outstanding photosynthetic parameters. *H. reticulata*, *V. mangachapoi*, *D. chunii*, and *U. sanyaensis* were clustered together with more similar leaf trait characteristics. The traits with large PCA1 factor loadings include eight leaf structural trait indicators, such as STT, DBAVB, and SAFVB, and nine photosynthetic parameter indicators, such as Pn, Pnmax, Rd, LCP, Tr, Gs, etc. PCA1 indicated the shift from anatomical dominance to photosynthetic trait dominance of the dominant species from left to right, and also represented the transition from shade-loving to sun-loving of the dominant species. The traits with large PCA2 factor loadings included six photosynthetic parameter indicators, Ci, WUE, Ls, Tr, Gs, and LSP and seven anatomical structure indicators, including DSVB, SASVB, SAFVB, SL, and SW. Most of these indicators with large factor loadings were related to water use, and PCA2 reflected the differences in water use ability of dominant species from top to bottom.

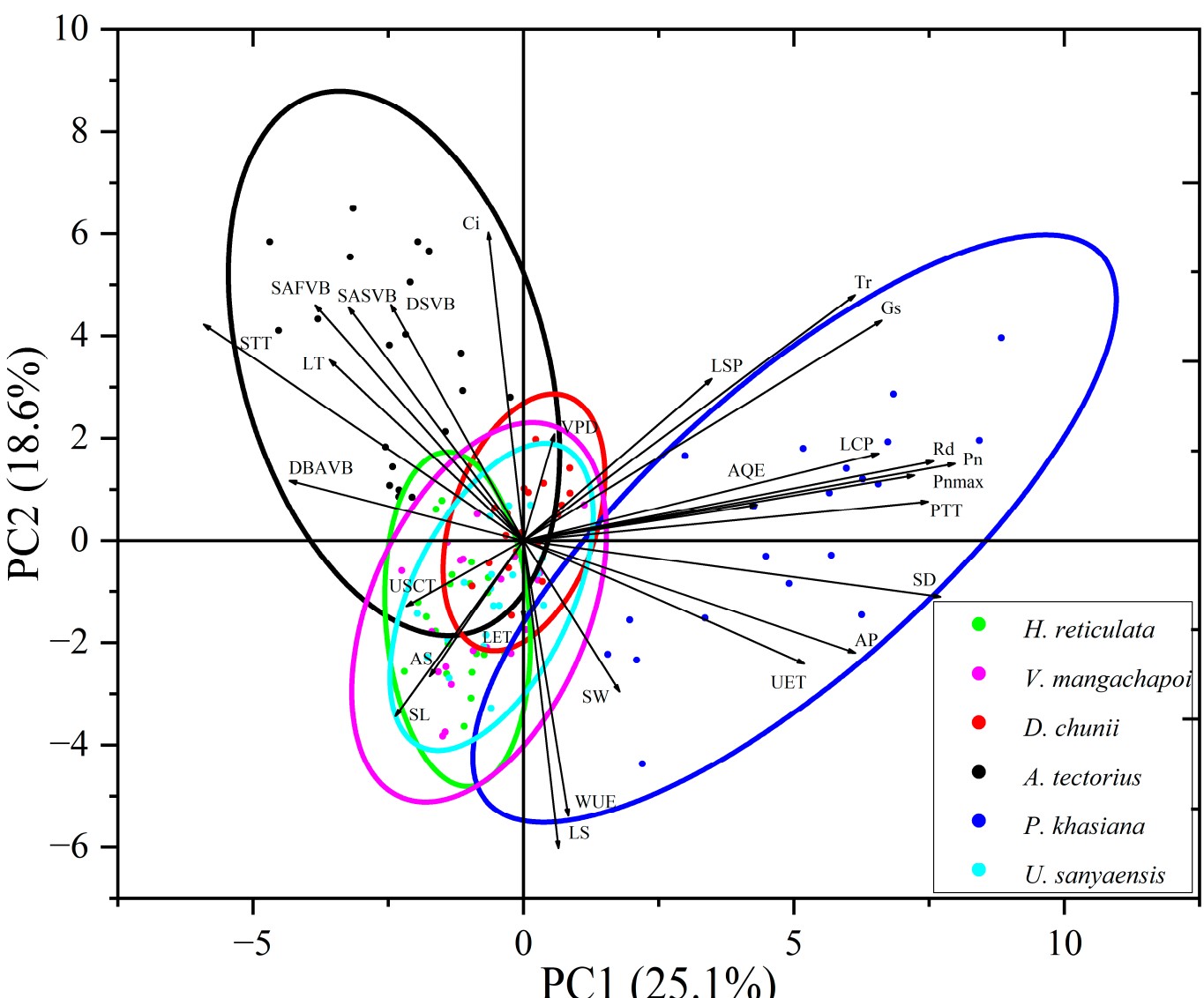

**Figure 6.** Principal component analysis of the leaf traits of the six dominant species. See Table 1 for full explanations of the abbreviations.

## 4. Discussion

### 4.1. Seasonal Variation of Leaf Functional Traits

A noticeable difference was observed in monthly gas exchange parameters, including Pn, Gs, Ci, and Tr, and light response curve characteristics, such as AQE and Pnmax, for dominant tree and vine species in tropical rainforests, and these photosynthetic physiological parameters (except for WUE) were significantly higher during rainy seasons, which are favorable for growth, which is consistent with the results of a photosynthetic characterization of tropical plants [26,27]. The seasonal variations in these photosynthetic parameters can be explained by seasonal changes in plant-accessible water, light radiation, and temperature; seasonal changes in photosynthetic performance were associated with the accumulation of leaf pigments (chlorophyll, carotenoids) and biochemical changes (rubisco concentration) or damage to photosystem II by drought [28–30]. In contrast to the monthly dynamics of photosynthetic physiological parameters, structural parameters such as leaf epidermal tissue, leaf flesh tissue, and vascular tissue differed less between months, and there were certain monthly differences only in stomata, but the monthly differences of most indicators were not significant ($p > 0.05$). The parameters of photosynthetic physiological characteristics such as Tr, WUE, LCP, and Rd of the dominant species in tropical rainforests were more plastic and had a higher ability to adapt to the environment. The lower plasticity of structural parameters reflects that the dominant species in tropical rainforests adapt to seasonal changes mainly by regulating photosynthetic physiological parameters and some stomatal parameters, but not the leaf tissue structure. Leaf tissue structure is the result of long-term adaptation of plants to their environment [31,32] and is relatively stable in the short term, with changes usually associated with drastic changes in leaf ecotype, growth stage, and environmental factors (temperature) [33,34], and tropical rainforests are characterized by evergreen species whose leaves do not shed annually, which indicates that these trees have adapted to their habitat over a long period of time.

### 4.2. Differences in Seasonal Drought Adaptation Strategies among Dominant Species

There are obvious seasonal droughts in the tropical lowland rainforest area of Ganzaling, Hainan, where rainfall in the rainy season accounts for more than 90% of the year, and the rainfall variation-induced change of rainy and dry seasons is an important environmental change in this area [35]. Tropical rainforests exhibit obvious seasonal variations in photosynthetic physiological parameters, as well as interspecific differences. Among the 15 leaf structural parameters, the maximum values of 12 structural parameters, including LT, USCT, and UET, were concentrated in three dominant vine species, and these parameters were correlated with plant drought tolerance [36–38], suggesting that dominant vine species have drought-tolerant leaf structures that enable them to cope with seasonal drought. Comparatively, the three dominant tree species, *H. reticulata*, *V. mangachapoi*, and *D. chunii*, displayed fewer advantages in drought tolerance characteristics, such as Pn, Gs, AQE, LT, PPT, and SAFVB, yet they had higher WUE and LDMC and smaller LA, indicating that the dominant tree species are capable of adapting to seasonal drought by enhancing their WUE and altering leaf morphology.

In seasonally dry tropical forests, trees grow mainly in the rainy season and largely stop growing in the dry season; the canopy vine, on the contrary, accounts for most of its annual growth in the dry season, which favors canopy vine dominance in the community [39]. *P. khasiana*, a canopy vine species, has eight gas exchange parameters, including Pn, Tr, and LSP. *P. khasiana* is a canopy vine with the highest eight gas exchange parameters and fitting parameters among the six dominant species, reflecting its strong photosynthetic capacity and sun-loving characteristics, as well as its thick fenestrated tissue thickness (PTT) and upper epidermal (UET), large and dense stomata, and densely distributed vascular bundles, and its strong leaf photosynthetic capacity and drought-tolerant tissue structure help it adapt to dry season stress and ensure its dominance in the community. This study reveals the mechanism of the community dominance of *P. khasiana* in seasonally dry tropical rainforests in terms of leaf photosynthetic physiology and anatomical structure. *A. tectorius*

and *U. sanyaensis* are interstratified plants, but they rarely reach the forest canopy, and their photosynthetic physiological and structural parameters such as leaf light saturation point (LSP), apparent quantum efficiency (AQE), fenestrated tissue thickness (PTT), stomatal density (SD) and stomatal apparatus area percentage (AP) are lower than those of the canopy vine *P. khasiana*. *A. tectorius* seems quite exploitative as it shows relatively high transpiration and low WUE compared to others species. Under drought conditions, *Carpinus betulus* in the understory of European floodplain forests also maximizes transpiration, stealing water from shallow-rooted tree species in the community and limiting their growth [40]. Whether *A. tectorius* also limits the growth of community shallow-rooted species such as *Carpinus betulus* needs further study. Three tree species (*H. reticulata*, *V. mangachapoi*, and *D. chunii*) showed more moderate shade tolerance, which is related to the selection of species by community succession, with shade-tolerant species being more dominant in the middle and later stages of succession [41]. Shade-tolerant species have higher hydraulic security than sunny species, are more adapted to seasonal drought [42], and are more competitive in seasonally dry tropical rainforests.

*4.3. Limitations and Future Research*

In this study, the photosynthetic properties, anatomical structure, and morphological characteristics of the leaves of six dominant species were studied in tropical lowland rainforest trees and vines on Hainan Island. However, the number of plant species selected in this study is relatively limited, with only three tree species and three vine species, and a lack of shrub species, which limits the ability to reveal the adaptation strategies of tropical lowland rainforest plants. To make the results more representative, subsequent studies should expand the number of plant species and investigate the hydraulic traits related to drought tolerance in plants. Thus, tropical lowland rainforest plants' adaptation strategies to droughts can be comprehensively understood. Additionally, only five individuals of each tree species were selected after considering policy restrictions and measurement workload. Future research will select a greater number of samples in order to verify the main findings of the present study.

**5. Conclusions**

Leaf photosynthetic physiological parameters display obvious monthly variation between the dominant species in tropical rainforests; however, leaf morphological and structural parameters show little variation. The physiological and structural parameters of photosynthetic processes and leaf morphology differ greatly between species. There is considerable plasticity in the leaf photosynthetic physiological parameters of tropical lowland rainforest dominant species, but less plasticity in leaf anatomical structure and morphology, suggesting that they adapt to seasonal changes mainly by regulating photosynthetic physiological parameters rather than leaf morphology and structure. The dominant tree species had greater WUE and LDMC than the dominant vine species, as well as a smaller leaf area (LA) than the dominant vine species, indicating that the tree dominant species relied on high water use efficiency and leaf morphological characteristics to adapt to seasonal changes. Three dominant vine species were found to possess the maximum values of 12 of the 15 leaf anatomical and structural parameters associated with drought tolerance, including LT, USCT, and UET, demonstrating that the vine plants adapted to dry season stress primarily through altering leaf anatomy structure related to drought tolerance. These results can provide a scientific basis for the conservation of tropical rainforests by illustrating the reasons for the establishment of dominant species in the community. Additionally, it would be beneficial to include more plant species and hydraulic trait parameters in future studies on seasonal adaptation strategies among dominant species in tropical rainforests, which will provide a comprehensive understanding of the adaptation strategies of tropical lowland rainforest plants to seasonal drought conditions.

**Supplementary Materials:** The following supporting information can be downloaded at: https://www.mdpi.com/article/10.3390/f14030522/s1, Table S1: Analytical parameters of leaf anatomical structure in six dominant species during different seasons.

**Author Contributions:** Conceptualization, R.X., S.F. and G.L.; methodology, R.X.; software, Q.Q. and J.N.; validation, R.X., G.L. and S.F.; formal analysis, Q.Q.; investigation, R.X.; data curation, R.X. and Q.Q.; writing—original draft preparation, R.X. and Q.Q.; writing—review and editing, G.L.; funding acquisition, R.X. and G.L. All authors have read and agreed to the published version of the manuscript.

**Funding:** This research was funded by the Fundamental Research Funds of ICBR Sanya Research Base (No. 1630032022005), and the Fundamental Research Funds for the International Centre for Bamboo and Rattan (No. 1632021021).

**Institutional Review Board Statement:** Not applicable.

**Informed Consent Statement:** Not applicable.

**Data Availability Statement:** All datasets presented in this study can be found within the article and in the Supplementary Materials.

**Acknowledgments:** The author would like to thank the Bamboo and Rattan Associated Tropical Rain Forest National ecological research station of Sanya, Hainan for its support to the field work of this study.

**Conflicts of Interest:** The authors declare no conflict of interest.

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
