# Peer review of "Seasonal Patterns and Species Variability in the Leaf Traits of Dominant Plants in the Tropical Rainforests of Hainan Island, China"

_forests, doi:10.3390/f14030522_

Round 1
Reviewer 1 Report
The manuscript describes and discusses analysis based on leaf physiological and anatomical to elucidate the effects of forest environment on different tree species.
This is a nicely written and potentially interesting study. However, if I understand the method description correctly, there was only 5 tree individual of each species assessed in the treatment group. This is totally insufficient to draw any conclusions that would be interesting for the readership. All these variables need to be taken into consideration by including sufficient number of biological replicates (= individual trees), exploring variability between these individuals and statistical significance of the differences both WITHIN and BETWEEN the experimental groups. Otherwise the research turns into a study of peculiar individuals, and drawing any general conclusions becomes impossible. For instance, what if a V. mangachapoi individual grew in the soil spot with altered concentration of phosphorus? Or what if the H. reticulata tree individual had a rare mutation that affects leaf anatomy? To account for these concerns, please include more tree individuals in the experimental group for the future submissions. Some corrections are detailed below
1. The Tropical Lowland Evergreen Forest is often known as the Neotropical Rainforest. This forest type is characterized by little or no seasonal water shortage and rather uniform warm and humid conditions. In such conditions the present experiment is not relevant.
2. whether a number of samples and selection of samples were statistically adequate e.g. correlation between parameters to draw reliable conclusions on the physiological properties of plant leaves with extreme diverse features.
3. Please add more supplemental figures with photos of all other tested trees from the experimental group, together with their GPS coordinates.
4. The tree leaves vary in age, in the position on a crown (geographical side, depth inside the canopy, height above the earth, etc.), which makes their physiology extremely diverse. To account for this, please state weather and how the leaves were pre-selected for your analyses within individual tree crowns. If there was such pre-selection, please state on which basis was it made. For example, were the leaves selected based on their similar age / developmental stage, depth inside the canopy, height above the earth or distance to the streetlamp. Please describe these pre-selection strategies in Materials and Methods.
5. It would be better to repeat the sampling, with a greater number of repetitions, to confirm the results obtained.
6. Hypothesis and objectives of the presented work is lacking in the introduction part.
Author Response
Dear reviewer,
Thank you for your letter and for the reviewer’s comments concerning our manuscript entitled “Seasonal patterns and species variability in leaf traits of dominant plants in tropical rainforests of Hainan Island, China”. Those comments are all valuable and very helpful for revising and improving our paper, as well as the important guiding significance to our research. We have studied comments carefully and have made correction which we hope to meet with approval. Revised portions are marked in red in the paper. The main corrections in the paper and the responds to the reviewer’s comments are as following point to point:
Point 1: There was only 5 tree individual of each species assessed in the treatment group. This is totally insufficient to draw any conclusions that would be interesting for the readership.
Response 1: Due to the time consuming measurement of photosynthetic parameters and other indicators in the field, the increase in the number of individuals makes the workload exponentially higher, and in addition, there are policy restrictions on conducting experiments in nature reserves. Therefore, in this study, five individuals of each tree species were selected after considering various factors. In the future, we will continue to expand the number of individuals to verify the results. This issue has been pointed out in the section of “4.3. Limitations and future research”. i.e. “Additionally, only five individuals of each tree species were selected after considering policy restrictions and measurement workload. Future research will select a greater number of samples in order to verify the main findings found in the present study.”
Point 2: The Tropical Lowland Evergreen Forest is often known as the Neotropical Rainforest. This forest type is characterized by little or no seasonal water shortage and rather uniform warm and humid conditions. In such conditions the present experiment is not relevant.
Response 2: Thanks to the very helpful suggestions of the reviewers, we conducted a check of the relevant publications and climate data. Seasonal drought is clearly present in the region (Figure 2 of the manuscript) and is also described in the published literature for tropical lowland rainforests (see [1], [2] in the Appendix). However, it is possible that we are not sufficiently aware of this, and to avoid misunderstanding, we have modified the full text from "tropical lowland rainforests" to "tropical rainforests".
Related published literature
[1]Hu, X.; Shu, Q.; Shang, Z.;Guo, W.; Qi, L. Secondary Succession in the Tropical Lowland Rainforest Reduced the Stochasticity of Soil Bacterial Communities Through the Stability of Plant Communities. Forests 2022, 13, 348. https://doi.org/10.3390/f13020348 .
[2]Zhao, Y.; LUAN, J.W.; Wang, Y.; Yang, H.; Liu, S.R. Effects of Simulated Drought and Phosphorus Addition on Nitrogen Mineralization in Tropical Lowland Rain Forests. Chinese Journal of Plant Ecology 2022, 46, 102–113, doi:10.17521/cjpe.2021.0191. in Chinese).
Point 3: whether a number of samples and selection of samples were statistically adequate e.g. correlation between parameters to draw reliable conclusions on the physiological properties of plant leaves with extreme diverse features.
Response 3: Thanks to the reviewer’s valuable comments, we will increase the sample size and expand the sample selection to validate the results of this study in future studies.
Point 4: Please add more supplemental figures with photos of all other tested trees from the experimental group, together with their GPS coordinates.
Response 4: Thanks for reviewer’s suggestion. To ensure that the habitats of the six dominant species were as similar as possible, we selected all test samples in similar sample plots (within a radius of 80 m). Due to the proximity of the test samples, the GPS coordinates of all samples were not recorded.
Point 5: The tree leaves vary in age, in the position on a crown (geographical side, depth inside the canopy, height above the earth, etc.), which makes their physiology extremely diverse. To account for this, please state weather and how the leaves were pre-selected for your analyses within individual tree crowns. If there was such pre-selection, please state on which basis was it made. For example, were the leaves selected based on their similar age / developmental stage, depth inside the canopy, height above the earth or distance to the streetlamp. Please describe these pre-selection strategies in Materials and Methods.
Response 5: The pre-selection strategy of the leaves is described in "2.3. Experimental design and sampling" of the manuscript. “Photosynthetic parameters were measured on a sunny day, and the youngest branch at 1.6 m above ground level was selected at five locations in the canopy: east, west, south, north, and center; and the developed mature leaf (penultimate 4th leaf at the end of the branch) was selected on the branch. ”
Point 6: It would be better to repeat the sampling, with a greater number of repetitions, to confirm the results obtained.
Response 6: Thanks for reviewer’s advice. Due to workload constraints and nature reserve policy implications, we will increase the number of sample replicates in future studies to validate the results obtained.
Point 7: Hypothesis and objectives of the presented work is lacking in the introduction part.
Response 7: Thanks for reviewer’s suggestion. The hypothesis and objectives of this study have been added to the Introduction.“We hypothesized that (1) there are species and seasonal differences in leaf functional traits (photosynthetic physiology and morphological and structural parameters) of dominant species in tropical lowland rainforests; (2) photosynthetic physiological parameters are the important leaf trait variations through which dominant species adapt to seasonal changes in their environment; (3) there are differences in environmental adaptation strategies between trees and vines. The findings could contribute to the understanding of the survival and adaptation strategies of tropical plant species in rainforests, as well as provide a scientific basis for the conservation and use of rainforest plant resources.”
We tried our best to improve the manuscript and made some changes in the manuscript. These changes will not influence the content and framework of the paper. And here we did not list the changes but marked in red in revised paper. We appreciate for reviewer’s warm work earnestly, and hope that the correction will meet with approval. Once again, thank you very much for your comments and suggestions.

Reviewer 2 Report
Review
The paper by Xu et al., titled: “Seasonal patterns and species variability in leaf traits of dominant plants in tropical lowland rainforests of Hainan Island, China “, is original work, fitting to the scope of Forests journal. The paper explored interspecific variability and seasonality of functional traits among 6 plant species in tropical lowland rainforest. Authors analysed leaf anatomical and morphological traits coupled with gas-exchange measurements conducted between June of 2019 and March of 2020. The phenotypic plasticity of the traits was further calculated. The study expands general ecophysiological knowledge of abiotic stress resistance, phenotypic plasticity and adaptation of woody plant species.
The paper is overall well composed and easy to read. Abstract is concise and informs the reader about most important findings of the paper. Introduction gives overview of the problematic, but could be improved. Authors should formulate hypotheses and concrete objectives at the end of introduction. Materials and methods sections is clear and authors describe the used methods in great detail. Statistical analyses are adequate, authors could explore trade-offs / coordination between traits with correlation analysis or principal component analysis. Visual presentation of results is clear and informative. Authors could maybe transform the Figure 3 and Figure 4 to the same style as Figure 5, to improve the clarity. This is just a suggestion. Authors discuss their results extensively and compare their results with other studies. The discussion enhances the overall value of the paper. Conclusion reflects the finding and is a great summary of the paper. I am looking forward to other great works of the authors.
I recommend minor revision of the paper to Forests. Please find my comments below:
Abstract
Line 17-22: This is extremely long sentence. Please separate it into more readable parts.
Introduction
Line 46: Authors should mention also increased evaporative demand caused by increasing temperature due to climate change: https://doi.org/10.3390/w14193015. That is also one of the reasons why the investigation of the leaf functional traits is crucial for forest ecology and ecophysiology. We need to understand how will the tree species react to the higher VPD in the future (also in rainforests).
Line 56: Authors could mention that the investigated leaf functional traits (such as SL, SD, SLA) are adaptive traits in tree species and characterize the population/species drought resistance and/or water retention ability: https://doi.org/10.1016/j.plaphy.2020.11.043.
Line 70: This paragraph is too short as the main objective of the study is seasonality. Tree species can adjust both their photosynthetic performance and water use efficiency during the exceptionally warm/dry seasons.
Discussion
Line 338: Seasonal changes in photosynthetic performance could be also explained by accumulation of pigments (chlorophylls, carotenoids) and biochemical changes (Rubisco concentration) or damage to photosystem II under drought (https://doi.org/10.1071/FP20040).
Line 383: It is interesting that A. tectorius seems quite exploitative as it is showing relatively high transpiration and low WUE compared to others species. You could compare it to the exploitative nature of understory Carpinus betulus from European floodplain forests, where it maximizes transpiration even under drought conditions, “stealing” water from cohabitant Oak and Ash trees.
Author Response
Dear reviewer,
Thank you for your letter and for the reviewer’s comments concerning our manuscript entitled “Seasonal patterns and species variability in leaf traits of dominant plants in tropical rainforests of Hainan Island, China”. Those comments are all valuable and very helpful for revising and improving our paper, as well as the important guiding significance to our research. We have studied comments carefully and have made correction which we hope to meet with approval. Revised portions are marked in red in the paper. The main corrections in the paper and the responds to the reviewer’s comments are as following point to point:
Point 1: Authors should formulate hypotheses and concrete objectives at the end of introduction.
Response 1: Thanks for reviewer’s suggestion. The hypothesis and objectives of this study has been added to the Introduction. “We hypothesized that (1) there are species and seasonal differences in leaf functional traits (photosynthetic physiology and morphological and structural parameters) of dominant species in tropical rainforests; (2) photosynthetic physiological parameters are the important leaf trait variations through which dominant species adapt to seasonal changes in their environment; (3) there are differences in environmental adaptation strategies between trees and vines. The findings could contribute to the understanding of the survival and adaptation strategies of tropical plant species in rainforests, as well as provide a scientific basis for the conservation and use of rainforest plant resources.”.
Point 2: Authors could explore trade-offs / coordination between traits with correlation analysis or principal component analysis.
Response 2: Thanks for reviewer’s advice . Principal component analysis of leaf traits has been added in section 3.5 of the manuscript.
Point 3: Authors could maybe transform the Figure 3 and Figure 4 to the same style as Figure 5, to improve the clarity.
Response 3: As reviewer’s suggested that has converted the Fig. 3 and Fig. 4 to the same style as Fig. 5.
Point 4: Line 17-22: This is extremely long sentence. Please separate it into more readable parts.
Response 4: The sentence has been modified to“Six dominant species including three trees (Hopea reticulata, Vatica mangachapoi, and Diospyros chunii) and three vine plants (Ancistrocladus tectorius, Phanera khasiana, and Uvaria sanyaensis) in tropical lowland rainforest in Ganzaling Nature Reserve of Hainan province were selected as study objectives. The key leaf traits were studied by using paraffin section method, leaf epidermis segregation method, and Li-6400 portable photosynthesis system in June, September, December 2019, and March 2020.”.
Point 5: Line 46: Authors should mention also increased evaporative demand caused by increasing temperature due to climate change: https://doi.org/10.3390/w14193015. That is also one of the reasons why the investigation of the leaf functional traits is crucial for forest ecology and ecophysiology. We need to understand how will the tree species react to the higher VPD in the future (also in rainforests).
Response 5: Thanks to the reviewer for his valuable comments. The Introduction of this manuscript has been added as following. “Increased evaporative demand caused by increasing temperature due to climate change[3].”.
Point 6: Line 56: Authors could mention that the investigated leaf functional traits (such as SL, SD, SLA) are adaptive traits in tree species and characterize the population/species drought resistance and/or water retention ability: https://doi.org/10.1016/j.plaphy.2020.11.043.
Response 6: As suggested, the Introduction of this manuscript has been added as following. “leaf functional traits (such as SL, SD, SLA) are adaptive traits in tree species and characterize the population/species drought resistance and/or water retention ability[9].”.
Point 7: Line 70: This paragraph is too short as the main objective of the study is seasonality. Tree species can adjust both their photosynthetic performance and water use efficiency during the exceptionally warm/dry seasons.
Response 7: Thanks for reviewer’s advice, the results of the study on the effect of seasonal drought on plant growth has been added to this paragraph. “Seasonal drought has a significant impact on plant growth, water physiology, nutrient cycling, and gene expression [15-19]. For example, seasonal drought reduces the physiological functions of leaves of tropical tree species in Panama and affects stomatal conductance of tropical forest leaves, which in turn reduces photosynthesis and growth rate of plants [19]; seasonal drought also leads to a reduction in soil moisture, which affects transpiration in subtropical coniferous forests[17].”.
Point 8: Line 338: Seasonal changes in photosynthetic performance could be also explained by accumulation of pigments (chlorophylls, carotenoids) and biochemical changes (Rubisco concentration) or damage to photosystem II under drought (https://doi.org/10.1071/FP20040).
Response 8: Thanks for reviewer’s suggestion. The Discussion of this manuscript has been added as following. “seasonal changes in photosynthetic performance were associated with accumulation of leaf pigments (chlorophyll, carotenoids) and biochemical changes (Rubisco concentration) or damage to photosystem II by drought [29–31].”.
Point 9: Line 383: It is interesting that A. tectorius seems quite exploitative as it is showing relatively high transpiration and low WUE compared to others species. You could compare it to the exploitative nature of understory Carpinus betulus from European floodplain forests, where it maximizes transpiration even under drought conditions, “stealing” water from cohabitant Oak and Ash trees.
Response 9: Thank you for the reviewer’s advice. The Discussion of this manuscript has been added as following. “A. tectorius seems quite exploitative as it is showing relatively high transpiration and low WUE compared to others species. Under drought conditions, Carpinus betulus in the understory of European floodplain forests also maximizes transpiration, stealing water from shallow-rooted tree species in the community and limiting their growth[41].Whether A. tectorius also limits the growth of community shallow-rooted species like Carpinus betulus needs further study.”.
We tried our best to improve the manuscript and made some changes in the manuscript. These changes will not influence the content and framework of the paper. And here we did not list the changes but marked in red in revised paper. We appreciate for reviewer’s warm work earnestly, and hope that the correction will meet with approval. Once again, thank you very much for your comments and suggestions.

Reviewer 3 Report
The title is representative, but I would suggest a more concise version in the future.
The abstract is fine, but it does not illustrate very well the purpose of the paper, which I think is very important. I also think that the justification for the research (its importance) should be specified more clearly.
In lines 77 and 78, I would write in letters the number 3.
Line 87 it is Fig. 1
The rest of the work, introduction, methods and results are good organized. I also recommend a very short linguistic check.
In conclusion, it is an interesting work and it can be of interest in the current conditions.
Author Response
Dear reviewer,
Thank you for your letter and for the reviewer’s comments concerning our manuscript entitled “Seasonal patterns and species variability in leaf traits of dominant plants in tropical rainforests of Hainan Island, China”. Those comments are all valuable and very helpful for revising and improving our paper, as well as the important guiding significance to our research. We have studied comments carefully and have made correction which we hope to meet with approval. Revised portions are marked in red in the paper. The main corrections in the paper and the responds to the reviewer’s comments are as following point to point:
Point 1: The title is representative, but I would suggest a more concise version in the future.
Response 1: Thanks to the reviewer 's suggestion. We have revised the title to “Seasonal patterns and species variability in leaf traits of dominant plants in tropical rainforests of Hainan Island, China”.
Point 2: The abstract is fine, but it does not illustrate very well the purpose of the paper, which I think is very important. I also think that the justification for the research (its importance) should be specified more clearly.
Response 2: Thanks for reviewer’s suggestion. The purpose and significance of this study has been added to the Abstract. “To investigate the dynamics of functional traits of dominant species in tropical rainforests and the differences in their adaptation strategies to seasonal drought, the results of the study can provide a scientific basis for tropical rainforest conservation resource protection. ”.
Point 3: In lines 77 and 78, I would write in letters the number 3.
Response 3: Thanks for reviewer’s suggestion, we have modified number 3 to the letter.
Point 4: Line 87 it is Fig. 1
Response 4: We are very sorry for our incorrect writing and we have corrected the full text where there were similar errors.
Point 5: The rest of the work, introduction, methods and results are good organized. I also recommend a very short linguistic check.
Response 5: Thank you for the reviewer’s recognition of the paper. As suggested, we have checked the language of the full text.
We tried our best to improve the manuscript and made some changes in the manuscript. These changes will not influence the content and framework of the paper. And here we did not list the changes but marked in red in revised paper. We appreciate for reviewer’s warm work earnestly, and hope that the correction will meet with approval. Once again, thank you very much for your comments and suggestions.

Round 2
Reviewer 1 Report
Dear Author,
I am satisfied with the comments, they had revised all the comments. Kindly accept the research paper in its current form. I believe that it makes a significant contribution to the field of biological research. I would like to take this opportunity to provide a review of your paper. I look forward to seeing your work published in our journal and to the impact that it will have in the scientific community.
Thank you for your hard work and dedication to this research, and congratulations on this achievement.
Author Response
Dear reviewer,
We sincerely thank the reviewer for the high recognition of the paper. The very valuable suggestions made by the reviewer are very helpful for the revision and improvement of our paper. Once again, thank you for your suggestions and for your positive comments on the paper.